# COVID-19 Detection via Silicon Nanowire Field-Effect Transistor: Setup and Modeling of Its Function

**DOI:** 10.3390/nano12152638

**Published:** 2022-07-31

**Authors:** Asma Wasfi, Falah Awwad, Juri George Gelovani, Naser Qamhieh, Ahmad I. Ayesh

**Affiliations:** 1Department of Electrical Engineering, College of Engineering, United Arab Emirates University, Al Ain P.O. Box 15551, United Arab Emirates; 201180954@uaeu.ac.ae; 2Zayed Center for Health Sciences, United Arab Emirates University, Al Ain P.O. Box 15551, United Arab Emirates; 3College of Medicine and Health Sciences, United Arab Emirates University, Al Ain P.O. Box 15551, United Arab Emirates; jgelovani@uaeu.ac.ae; 4Department of Physics, College of Science, United Arab Emirates University, Al Ain P.O. Box 15551, United Arab Emirates; nqamhieh@uaeu.ac.ae; 5Physics Program, Department of Mathematics, Statistics and Physics, College of Arts and Science, Qatar University, Doha P.O. Box 2713, Qatar; ayesh@qu.edu.qa

**Keywords:** COVID-19, FET biosensor, semiempirical modeling

## Abstract

Biomolecular detection methods have evolved from simple chemical processes to laboratory sensors capable of acquiring accurate measurements of various biological components. Recently, silicon nanowire field-effect transistors (SiNW-FETs) have been drawing enormous interest due to their potential in the biomolecular sensing field. SiNW-FETs exhibit capabilities such as providing real-time, label-free, highly selective, and sensitive detection. It is highly critical to diagnose infectious diseases accurately to reduce the illness and death spread rate. In this work, a novel SiNW-FET sensor is designed using a semiempirical approach, and the electronic transport properties are studied to detect the COVID-19 spike protein. Various electronic transport properties such as transmission spectrum, conductance, and electronic current are investigated by a semiempirical modeling that is combined with a nonequilibrium Green’s function. Moreover, the developed sensor selectivity is tested by studying the electronic transport properties for other viruses including influenza, rotavirus, and HIV. The results indicate that SiNW-FET can be utilized for accurate COVID-19 identification with high sensitivity and selectivity.

## 1. Introduction

For the last twenty years, various nanomaterials including nanogaps, nanotubes, nanowires, nanoparticles, and nanoscale films [1,2,3,4,5,6] have attracted researchers’ interest due to their potential for designing nanoscale sensors. Various nanoscale sensing methods have been utilized in biological applications and research. Precise and rapid detection mechanisms are required to monitor living systems. The major factors in designing and fabricating the biomolecular sensors are a low cost, quick and accurate results, and high sensitivity and selectivity. Field-effect transistors (FETs) have potential in sensor applications due to their ability to translate the molecule’s interaction with the sensor to readable signals in real time [7,8,9]. Recently, various semiconducting materials types such as nanowires [10,11] and carbon materials [12,13] have shown promise in the fabrication of field-effect transistor-based sensors. Carbon-based sensors have been developed for different applications such as glucose concentration detection [14], DNA hybridization [15,16], antigen–antibody interactions [10,17,18], and cancer biomarkers detection [19,20]. Despite the advantages of carbon-based sensors such as carbon nanotube (CNT) FETs in biomolecular applications, several limitations were identified in the fabrications processes and applications as well. The fabrication of CNT-FETs with both metallic and semiconducting elements still requires enhancements in nanoelectronics. The sensing techniques involving field-effects are complex [21,22], including Schottky barriers [23], and electron transfer [21]. On the other hand, the sensing mechanism of silicon nanowire field-effect transistor (SiNW-FET) sensors is straightforward [24,25] due to the direct interaction between the SiNW-FET and the target in solution.

The silicon industry is well-developed, with mature processing and fabrication methods. Such a mature industry enables the fabrication of SiNWs with different shapes [26], dopants [27], and sizes [28,29]. SiNWs performance shows a high reproducibility because it can be monitored during the wire growth.

Silicon is vastly utilized in modern microelectronic devices. Various theoretical and experimental works have been carried out to understand the properties and the structures of silicon clusters doped with transition-metals (TM) [30,31]. These clusters are useful for solar cells, lithium-ion batteries, silicon-based catalysts, and cluster assembled materials. Single TM-atom-doped silicon clusters have special geometric structures, novel electronic properties, and special magnetic characteristics such as ferromagnetism and anti-ferromagnetism. The electronic properties can be modified by varying the central metal atom [30,31].

SiNWs’ doping, mobility, and density can be determined in advance. It is highly important to detect infectious diseases accurately to monitor their spread rate and implement various methods of epidemiological control. One of the rapid virus detection techniques is reverse transcription-polymerase chain reaction (RT-PCR) which requires less than one day and provides a high sensitivity [32]. Moreover, the quick antigen detection test can be used for influenza virus diagnosis within 30 min, but this method has a lower sensitivity in comparison with RT-PCR. Biomolecular detection methods can be categorized based on their labeling requirements. 

All labeling techniques are considered expensive, cumbersome, and time-consuming [33]. Field-effect transistors have been attracting massive attention because of their label-free detection, real-time response, high sensitivity, and compact size [32,34,35]. Lately, various kinds of silicon-based sensors have been utilized for the various types of detection including viruses, proteins, DNA, and ions [36,37,38,39,40,41]. One of the essential limitations of the mentioned methods is the possibility of false positive results. In biological research, SiNW-FETs are utilized for various types of detection including viruses, cancer biomarkers, small molecules, and DNA sequence. Quick, cheap, and robust analytical methods that are capable of detecting viruses accurately are critical for the enhanced control and prevention of infections.

Coronavirus disease 2019 (COVID-19) is a rapidly spreading virus that causes an acute respiratory syndrome in humans and can cause death. COVID-19 infection was classified as a pandemic by the World Health Organization (WHO). The early diagnosis of COVID-19 is critical to contain the outbreak. COVID-19 consists of four structural proteins: spike protein, nucleocapsid, matrix, and envelope [42]. The spike protein is the target molecule since it decorates the exterior of the COVID-19 virus, and it is the first part that will interact with the biosensor. 

The overall aim of our current research was to develop a SiNW FET sensor for COVID-19 detection, study the electronic transport properties (i.e., transmission spectrum, conductance, and current) while interacting with COVID-19 in comparison with other types of viruses. The specific aim of the current simulation work was to assess the feasibility of a SiNW-FET biosensor for the detection of COVID-19 viral particles before proceeding to its fabrication.

The semiempirical modeling approach was used to capture the COVID-19 viral particles on the sensor surface. A specific antibody was used to detect the COVID-19 spike protein. We demonstrate that the SiNW-FET biosensor should be capable of detecting COVID-19 viral particles with high sensitivity.

This study is a proof of concept that the designed SiNW-FET can be used as a sensor for COVID-19 virus detection. This work was able to find a unique electronic signature and detect COVID-19 antigen using the designed sensor.

The novelty in this article is based on the use of the SiNW-FET for the first time as a sensor to detect the COVID-19 spike antigen. Moreover, a semiempirical model was used, which was an extension of the extended Hückel approach with a self-consistent Hartree potential. The semiempirical model has the advantage of a lower computational cost and the possibility to use it in parallel with an experiment where the model can be fitted to get accurate results [43].

## 2. Materials and Methods

### 2.1. Sensor Setup and Configuration

The Quantumwise Atomistix ToolKit (ATK) with the graphical user interface Virtual NanoLab (VNL) was used to set up the SiNW-FET [44]. The simulation and modeling of the SiNW-FET with a gate dielectric of silicon dioxide was generated.

Figure 1 illustrates the sensor configuration, which consisted of three regions, the source, drain, and channel, and the gate electrode placed underneath the channel. The silicon nanowire edges were saturated with hydrogen. The silicon nanowire was oriented in the (100) direction where the two ends of the nanowire were doped to get an n-type doping concentration of 4 *×* 10^19^ cm^−3^ in the source and drain regions. 

The gate was made of two layers: a dielectric layer of SiO_2_ with a relative dielectric K = 3.9 and a metallic layer. The metal layer can be made of palladium, platinum, iridium, or rhodium. However, ATK-VNL does not provide the option to select the metal type. The SiNW-FET channel had an approximate width of 57 Å and length of 75 Å. These dimensions were selected due to the number of atoms and the size of the COVID-19 target molecule. A pair of 12 Å electrodes was connected at the edges of the channel. 

Moreover, it is worth noting that QuantumATK provided computational details for nanoscale sensors, which could be used as a proof of concept to support the experiment based on bigger size sensors.

For the simulation, the Si crystal orientation (111) in the orientation of the nanowire was used. All atomic locations and the lattice constant were optimized. The reactive dangling bonds on silicon atoms were passivated with hydrogen to enhance the material’s stability and lifetime [45]. These dangling bonds can modify the bandgap energy of the material, which affects its semiconductor properties.

Figure 2 shows the SiNW-FET with the spike protein and a top view of the channel to study the variation on the electronic transport properties. The atomic coordinates and structure factors for the virus and antibody are taken from the RCSB Protein Data Bank [46,47].

In the simulation, the antibody–antigen binding structure was utilized. The PDB ID of (Spike protein) is 2IEQ [46] while the PDB ID of (COVID-19 virus spike receptor-binding domain complexed with a neutralizing antibody) is 7BZ5 [47]. The second structure consisted of neutralizing antibodies binding to the spike glycoprotein receptor of COVID-19 virus. Moreover, the SiNW-FET selectivity was studied where the whole virus was used for HIV, rotavirus, and influenza. The PDB ID for each of the viruses is as the following: for HIV, 4XFZ [48]; for rotavirus, 3MIW [49]; and for the influenza virus,1EA3 [50].

The concentration of the antibody and antigen on the SiNW-FET was kept low (one molecule) to reduce the required time to conduct the simulation. Although a low concentration of the target molecules was used in the simulation, each one of the IV curves took more than three weeks to generate with HPC. The current variation and sensitivity was expected to be more obvious with a higher concentration of the target molecules as shown in our previous work [51].

Figure 3 shows the SiNW-FET coated with an antibody against the spike protein of the COVID-19 virus to detect the binding of the spike protein to the antibody by analyzing the variation in the electronic transport characteristics. The spike protein separately and the spike protein bound to the spike antibody were detected via SiNW-FET. During the electronic transport calculation of the simulated biosensor, a 1 V gate potential was fixed and a DC bias voltage of 0.1 V, 0.2 V, 0.3 V, and 0.4 V was applied to the source and drain.

This study was a proof of concept that the designed SiNW-FET could be used as a sensor for COVID-19 virus detection. The antibody was only used with the COVID-19 spike target and it was not added for the other viruses such as HIV and rotavirus due to the complexity of calculations and time required. The results indicated that the sensor showed the highest variation in current due to the COVID-19 spike protein antigen in comparison to the other types of viruses such as HIV, rotavirus, and influenza virus. Moreover, the variation in current due to the binding between the spike protein antigen and antibody of the COVID-19 virus was studied. In the experiment, the COVID-19 spike antibody was used to ensure the sensor selectivity to the COVID-19 spike antigen [42,52,53]. It was expected that the target molecules of COVID-19 spike antigen would bind to the COVID-19 spike antibodies, whereas the other viruses would not bind to the antibodies.

### 2.2. Computational Methodology

In this work, ATK with VNL was used to design the SiNW-FET sensor and to conduct the study. ATK-VNL has various built-in calculators which use various methods to generate the electronic parameters of the simulated sensors. The simulation work was performed by utilizing a semiempirical modeling. 

To generate the biosensor electronic transport characteristics, NEGF+SC-EH simulations were performed via the ATK-SE semiempirical model. The utilized semiempirical model is an extension of the extended Hückel method [43]. The sensor performance was analyzed using the nonequilibrium Green’s function combined with the self-consistent extended Hückel (NEGF+SC-EH) method. Several electronic properties were studied including conductance, transmission spectrum, and the electrical current of the sensor before and after placing COVID-19 on its surface. This study produced a unique current for each virus placed on the sensors’ surface.

A high-performance computing (HPC) environment was used to speed up the simulation process. Seven computing nodes were used, each one having 36 cores. In total 252 cores were used to conduct the simulation. The ATK-VNL workflow started by using the ATK builder tool to form the device structure. Then, the designed structure was generated as a python script via the script generator. The script generator enabled the addition of an inbuilt calculator with the required parameters to analyze the nanodevice. The scripts were generated as a python code that could be modified and edited to conduct the required simulations. In the last step, the python script was sent to the job manager to be run on the HPC environment. The output files generated were analyzed and viewed.

The various electronic transport characteristics of the sensor can be generated, after obtaining the self-consistent nonequilibrium density matrix. The transmission spectrum is one of the most notable transport properties of the system from which the current and conductance are calculated. The transmission coefficient T at electron energy can be obtained from the retarded Green’s function [54].

The zero-bias transmission spectrum between the source and the drain was calculated using Equation (1) [55,56]:(1)$$\;T\left(E\right)=Tr\left\{\Gamma_{D}\left(E\right)G\left(E\right)\Gamma_{S}\left(E\right)G^{†}\left(E\right)\right\}$$
where E is the energy,  Tr is the trace, $\Gamma_{D,\;S}\left(E\right)=i\left[\sum_{L,\;R}\left(E\right)-\sum_{S,\;D}^{†}\left(E\right)\right]$ describes the broadening level because of the coupling to the electrodes, and $\sum_{L,\;R}\left(E\right),\;\sum_{S,\;D}^{†}\left(E\right)\;$ are the self-energies presented by the electrodes.

The charge transport properties were computed by the semiempirical model based on SE-EHT [57] to acquire valid results. The semiempirical model has the advantage of a low computational cost and low computational time [58]. The two main quantities that illustrate the electron transport efficiency were the conductance and current, which were proportional to the transmission probability between the source and the drain based on the Landauer formula [59]. The total current was expressed by Equation (2) [56]:(2)$$I=\frac{2e}{h}\overset{\infty }{\underset{-\infty }{\int }}T\left(E,V_{b}\right)\left[f_{L}\left(E-{µ}_{L}\right)-f_{R}\left(E-{µ}_{R}\right)\right]\;dE.\;\;$$

E refers to the energy, $T\left(E,V_{b}\right)$ refers to the transmission matrix which could be generated from the molecular energy levels and their coupling to metallic leads. $f_{L}$ refers to left electrode Fermi function, while $f_{R}$ refers to the right electrode Fermi function. E refers to the energy in the conducting level, ${µ}_{L}$ and ${µ}_{R}$ are the energy potentials released or absorbed during the electron transition phase or chemical potential at the left and right electrodes, and the factor 2 relates to the spin degeneracy.

The conductance could be calculated from the transmission spectra as shown in Equation (3) [55,56]:(3)$$G=\frac{2e^{2}}{h}T$$
where T refers to the overall transmission probability which is the summation of all possible transmission channels. h refers to Planck’s constant, and e refers to the electron charge.

A geometry optimization was performed before generating the electronic transport properties for the developed sensor. The density functional theory was employed for the geometry optimization where a Perdew–Burke–Ernzerhof parameterization of the generalized gradient approximation was used. The cut-off energies were set to 125 Ha and the k-points sampling were set to 1 × 1 × 11.

To generate the biosensor electronic transport characteristics, NEGF+SC-EH simulations were performed via the ATK-SE package. To take into consideration the applied bias voltage effect, the SC Hartree potential was used [43] in the conventional EH model. The NEGF+SC-EH model in the ATK-SE package was illustrated in [43]. The Fermi level of the left and right electrodes was generated self-consistently. The grid mesh cut-off energy was 20 Hartree, while 2 × 2 × 50 k-points were selected. The Poisson equation was employed, where the Dirichlet boundary condition was used in the C direction and the Neumann boundary conditions were employed in the A and B directions [60]. A, B, and C are indictors for the A-, B- and C-direction as displayed in Figure 1b.

### 2.3. Sensing Methodology

Figure 4a shows the SiNW-FET which consists of three terminals (source, drain, and gate) and a semiconducting channel. The channel connects the source and drain electrodes while the gate modulates the conductance via an applied electrical potential. The sensing element of the SiNW-FET is the channel that alters the current due to the placement of the target molecule. The SiNW-FET has a recognition molecule (spike antibody) attached to the channel surface as displayed in Figure 4b, which specifically detects the target molecule (spike protein) as displayed in Figure 4c leading to variation in the sensor current as displayed in Figure 4d. Before generating the current, it is highly critical to modify the sensor surface to enhance the nanodevice sensitivity. The SiNW-FET channel was modified and coated with the COVID-19 spike protein antibody.

Figure 5 displays the working principle of the developed sensor. Figure 5a displays the n-type SiNW-FET where the electrodes are n-doped, while the channel is semiconducting. The COVID-19 spike protein antibody is immobilized to the SiNW-FET channel to capture the target COVID-19 spike protein molecule. Figure 4b shows that when the negatively charged target molecule binds the n-type SiNW-FET biosensor, a depletion of the charge carriers occurs leading to a decrement in the electrical current.

## 3. Results

To investigate the selective identification of the COVID-19 spike protein using the SiNW-FET biosensor, the COVID-19 spike protein antibody was immobilized on the SiNW-FET channel. The current was then generated for the SiNW-FET with and without the COVID-19 spike protein. The electronic transport behavior was investigated as below.

### 3.1. Transmission Spectrum

The transmission coefficients T(E) was generated for an energy range between −2 and 2 eV, and four biases V = 0.1, 0.2, 0.3, and 0.4 V at a 1 V gate potential, with results presented in Figure 6. The energy domain −2 to 2 eV has 200 sampling points. For the Hückel basis set, the empirical potentials “Cerda. Silicon [GW diamond]” basis sets were chosen for silicon. They were fitted to GW calculations, which gave an excellent description of the silicon band structure including the value of the bandgap.

The transmission spectra at different biases are displayed in Figure 6 as a function of the electronic energy for the SiNW-FET with and without the required molecules. The green curves show the SiNW-FET transmission spectra while the red curves show the SiNW-FET transmission spectra due to the COVID-19 spike protein (Figure 6a) and due to the bounding between the spike antibody and spike protein (Figure 6b). The SiNW-FET has enhanced transmission for the charge carriers before placing the target molecules. The spike protein is negatively charged, which results in the depletion of the charge carriers for the SiNW-FET.

Moreover, the transmission spectra show the central semiconducting SiNW-FET channel where the transmission has low values between −1.4 and 0.02 eV of energy due to the absence of energy levels within this area. The change in the transmission spectra due to the addition of the COVID-19 spike protein has changed the total electric current of the SiNW-FET.

### 3.2. Conductance and Current

The presence of the spike protein can be detected as a function of the change in conductivity and current of the SiNW-FET biosensor. Th SiNW-FET sensor was tested for different types of viruses. The variation in the current was used to evaluate the sensor performance. The change in the current was calculated as the current for the sensor due to the placement of the target molecule minus the current of the bare sensor. The drop of the drain current of the SiNW-FET sensor after placing the target molecule was due to the negatively charged spike protein, which induced excess hole carriers. Figure 7a reveals that the bare SiNW-FET has higher conductance than the SiNW-FET + COVID-19 antigen or the SiNW-FET + COVID-19 antibody + COVID-19 antigen. The addition of the antigen or antigen and antibody together modifies the surface of the SiNW-FET resulting in a conductance reduction. The decrement in the SiNW-FET mobility occurred due to additional sources of dispersion of the charge carrier. This variation in conductance results in a unique signature due to the physical and chemical structures of the target molecule. The COVID-19 spike protein is negatively charged, which binds with the antibody resulting in a decrement in the conductance and current response. SiNW-FET shows a maximum conductivity at 0.2 V. Thereafter, the conductivity gets reduced. The reduction in conductance at 0.3 V is due to the negative differential resistance. Figure 7a shows the conductance while Figure 7b shows the current–voltage curves for the SiNW-FET at 0.1, 0.2, 0.3, and 0.4 V before and after placing the COVID-19 spike protein. The variation in current due to the COVID-19 placement on the biosensor channel indicates the successful detection of the COVID-19 virus. The adsorbed spike protein interacts with the SiNW-FET channel and modifies its conductivity by modifying the carrier’s concentration.

Figure 7b shows the drain current (I_d_) versus the drain–source voltage (V_ds_) for the SiNW-FET at room temperature before and after placing the COVID-19 spike protein. The gate potential was fixed at 1 V while the V_ds_ was set to 0.1, 0.2, 0.3, and 0.4 V. The simulated sensor was an n-type transistor where fixing a positive gate potential increases the conduction of electrons within the transistor channel leading to an increment in the drain current.

A nonlinear resistance results in a nonlinear IV curve. Examples of passive nonlinear devices are diodes, transistors, and thyristors. The nonlinear relation is due to using silicon as the semiconducting material, which results in a nonlinear resistance leading to a nonlinear IV curve. Moreover, the reduction in conductance at 0.3 V caused by the negative differential resistance. 

Figure 8 shows the variation of current at V_ds_ = 0.1, 0.2 0.3, and 0.4 V at V_g_ = 1 V when the sensor was exposed to the COVID-19 spike protein. It is observed in Figure 7b that the current drops due to the placement of the COVID-19 spike protein. This variation of current is utilized to inspect the sensor performance. The current drop is due to the negatively charged spike protein, which induces excess hole carriers. Various studies have reported that the adsorption of the COVID-19 spike protein by the channel reduces the current since the holes concentration increases compared to the electrons [25,32,42]. Thus, the SiNW-FET electrical resistance increases due to holes trapping the electrons, leading to a current decrement. The difference in drain current (∆I) is larger when the bias voltage is increased. Figure 8 depicts that the ∆I increase is due to an increment of bias voltage. Moreover, Figure 8 shows that functionalizing the SiNW-FET channel with the COVID-19 antibody results in a higher variation in current and a higher sensitivity to the COVID-19 spike protein. 

Both the negative charge of the antibody and antigen and the adsorption of the molecules (antibody and antigen) lead to a unique variation in the electrical current of the SiNW-FET. Each molecule has a unique electronic state, size, and interaction with the SiNW-FET channel. Each molecule density of states contribution at the Fermi level is unique due to the different spatial extension of the molecule.

Previous experimental work showed that functionalizing the transistor with antibody resulted in a slight variation in current [42]. Thus, the simulation work did not include the results of SiNW FET + antibody, to avoid the computational cost. The effect of functionalizing the sensor with antibody was expected to be small, which would result in slight changes in the current [42]. The spike antibody was essentially added to ensure the selectivity of the sensor to COVID-19 spike antigen. Moreover, it resulted in a slight increment in current variation, which meant a higher sensitivity. Thus, the addition of the COVID-19 spike antibody resulted in a higher sensitivity and selectivity.

The field-effect transistor’s temperature-dependent conductance displayed in Figure 9 shows an initial decrease in conductance as a function of temperature. The conductance increases at the Fermi level due to the electron tunneling. At higher temperatures, the conductance increment starts stabilizing. Figure 9 shows that at 300 K and higher, the FET conductance is stable. 

In the designed sensor, the optimal results were generated when the bias potential was fixed among the left and right electrodes (+0.15 and −0.15 eV). Fixing the bias voltage at 0.3 V resulted in the best sensitivity readings. The orientation effect of the protein and of the antibody was expected to be very small and should not affect the readings. Each molecule had a unique chemical and physical structure, which affected the sensor current in a unique way resulting in a unique signature. This study was a proof of concept that the designed SiNW-FET could be used as a sensor for COVID-19 virus detection. The results were obtained by using the most common orientation where the antibody was on the channel surface and the antigen was bound to the antibody. This led to a distinctive current for each type of virus.

The SiNW-FET was tested against other viruses such as influenza, rotavirus and HIV to analyze its selectivity. The sensor’s response (variation in current) is displayed in Figure 10. The highest change in electrical signal was when the sensor was exposed to COVID-19. These results demonstrate a high selectivity for the SiNW-FET to COVID-19. The electrical potential and charge changes at the SiNW-FET surface are due to their binding or adsorption of charged target molecule, which alters the density of charge carriers in the channel. Thus, the channel current and conductivity among the source and drain change [7].

COVID-19 consists of four structural proteins: the envelope, spike, matrix, as well as the nucleocapsid. Among these four proteins, the spike protein is the best part to detect because it is highly immunogenic, and a major transmembrane protein [42]. Moreover, the spike protein enables the specific detection of the COVID-19 virus since it has amino-acid sequence diversity [42,61]. Thus, a spike antibody was designated as a receptor to detect the virus in this work. Receptor molecules (spike antibodies), immobilized on the SiNW(s), were used to recognize a particular target with a SiNW FET biosensor. The binding of the negatively charged spike protein antigen to spike antibodies changed the conductance of the semiconducting channel. For an n-type SiNW FET biosensor, the negatively charged target spike protein antigen produces a depletion of charge carriers within the whole cross-section of the device leading to a low source–drain current. The antibody molecule is immobilized to the channel surface and can selectively detect and capture the target spike protein antigen. The negatively charged spike protein antigen binds to the receptor (spike antibody) on the channel of the n-type sensor leading to a depletion of the charge carriers inside the SiNW, leading to a decrement in electrical current [24,25]. It is noticed that the current variation was slightly higher for the spike protein antigen bound to the spike antibody compared to the spike protein antigen only. The spike protein antigen bound to the spike antibody had a higher negative charge than the spike protein antigen only, resulting in a higher variation in current. This novel biosensor used the change in current, conductance, and transmission spectra to detect the target molecule. The simulation results showed significant variations in the current, conductance, and transmission spectra in the presence of the spike protein target molecule. These variations in the mentioned parameters were utilized as the identification signal to design a physical biosensor.

## 4. Conclusions

In this work, a silicon nanowire field-effect transistor (SiNW-FETs)-based biosensor was developed and functionalized with anti-COVID-19 spike protein antibody for the detection of COVID-19 viral particles using a semiempirical modeling approach. The SiNW-FET consisted of two electrodes, a channel, and a gate, where the charge transport properties were studied and analyzed. The simulation results showed a significant variation in the current, conductance, and transmission spectra in the presence of the COVID-19 spike protein as the target molecule. Moreover, the designed sensor should be able to differentiate the COVID-19 spike protein from other viruses such as influenza, HIV, and rotavirus. The results demonstrated that the SiNW-FET biosensor had the potential to successfully detect the COVID-19 virus.

## Figures and Tables

**Figure 1 nanomaterials-12-02638-f001:**
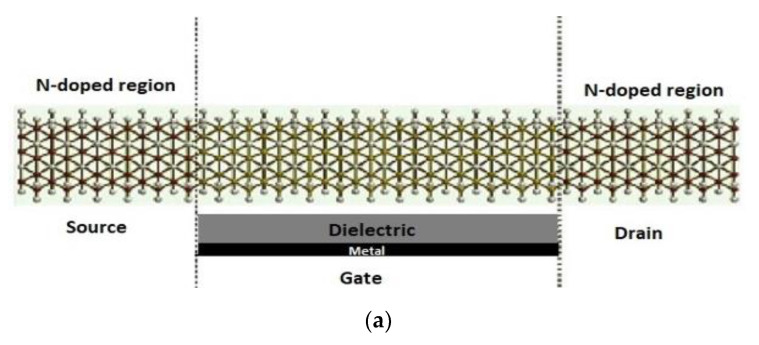
(**a**) Schematic representation of SiNW-FET composed of source, drain, channel, and gate underneath the channel. The source and drain are doped with n-type dopants and the gate consists of two layers: a dielectric layer and a metallic layer. (**b**) ATK-VNL view of the SiNW-FET. Silicon—yellow; hydrogen—white.

**Figure 2 nanomaterials-12-02638-f002:**
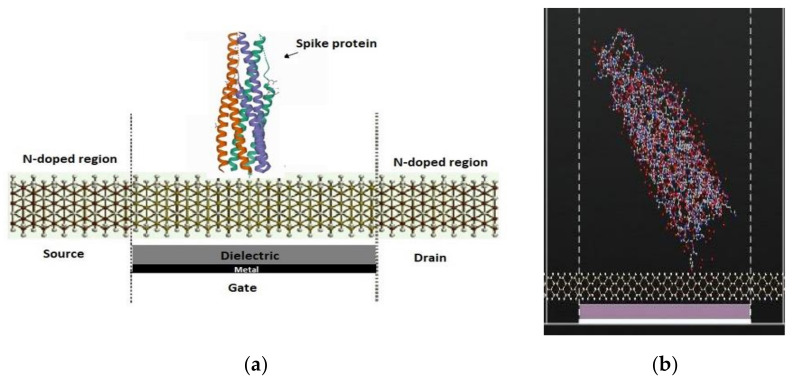
(**a**) Schematic diagram of the SiNW-FET for the detection of the spike protein. (**b**) ATK-VNL view of the SiNW-FET with spike protein placed on top of the channel.

**Figure 3 nanomaterials-12-02638-f003:**
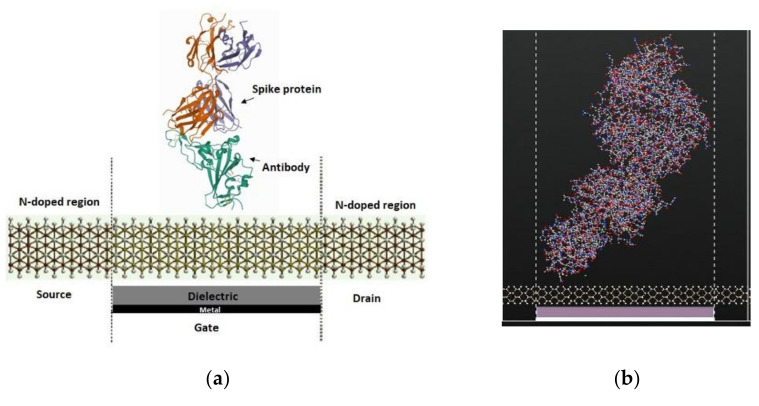
(**a**) Schematic diagram of the SiNW-FET coated with spike antibody for the detection of the spike protein. (**b**) ATK-VNL view of the SiNW-FET with a spike antibody bound to the spike protein.

**Figure 4 nanomaterials-12-02638-f004:**
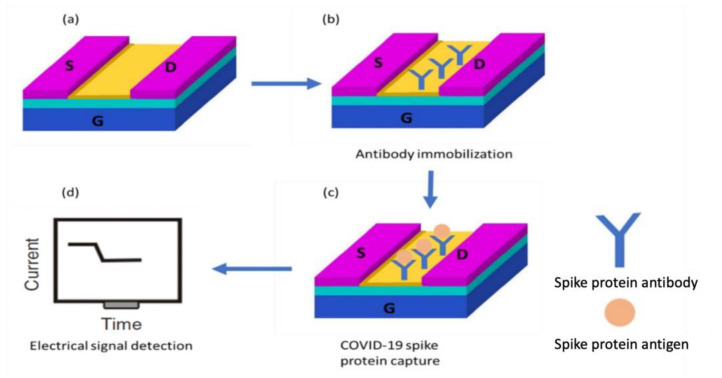
(**a**) Schematic diagram of the n-type SiNW-FET biosensor. (**b**) COVID-19 spike protein antibody immobilized on the SiNW-FET channel. (**c**) Target COVID-19 spike protein captured by the biosensor. (**d**) The current was generated for the SiNW-FET.

**Figure 5 nanomaterials-12-02638-f005:**
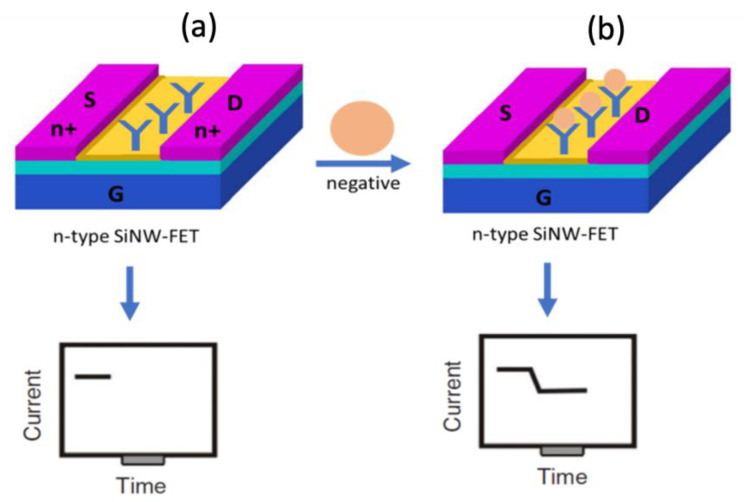
SiNW-FET biosensor working principle. (**a**) Schematic diagram of the SiNW-FET where a COVID-19 antibody was anchored to the channel. (**b**) Negatively charged spike protein captured by the biosensor leading to a current drop.

**Figure 6 nanomaterials-12-02638-f006:**
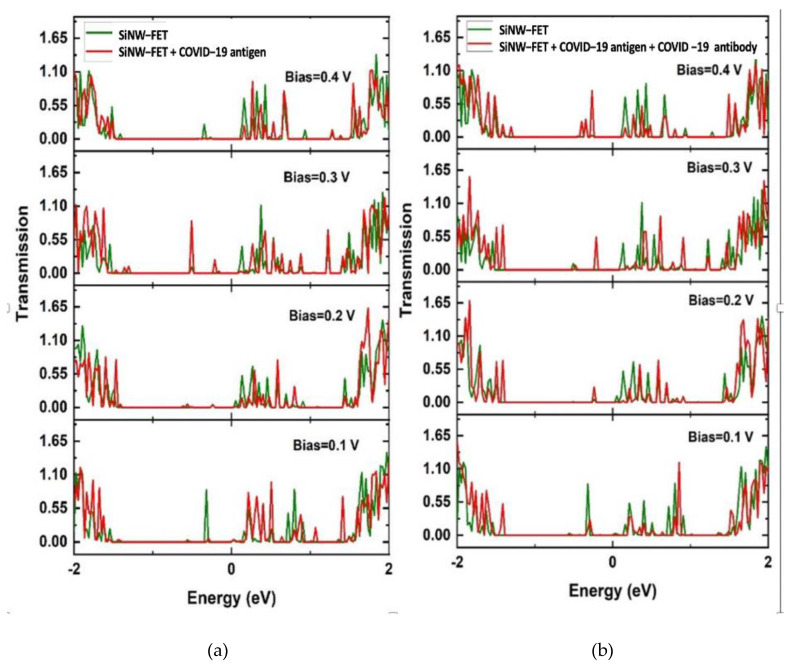
Transmission spectra T(E) versus bias V = 0.1, 0.2, 0.3, 0.4 V for SiNW-FET biosensor with two different molecules: (**a**) spike protein and (**b**) spike protein bound to a spike antibody. The green and red curves represent the electronic transmission through SiNW-FET and SiNW-FET with the spike protein, respectively.

**Figure 7 nanomaterials-12-02638-f007:**
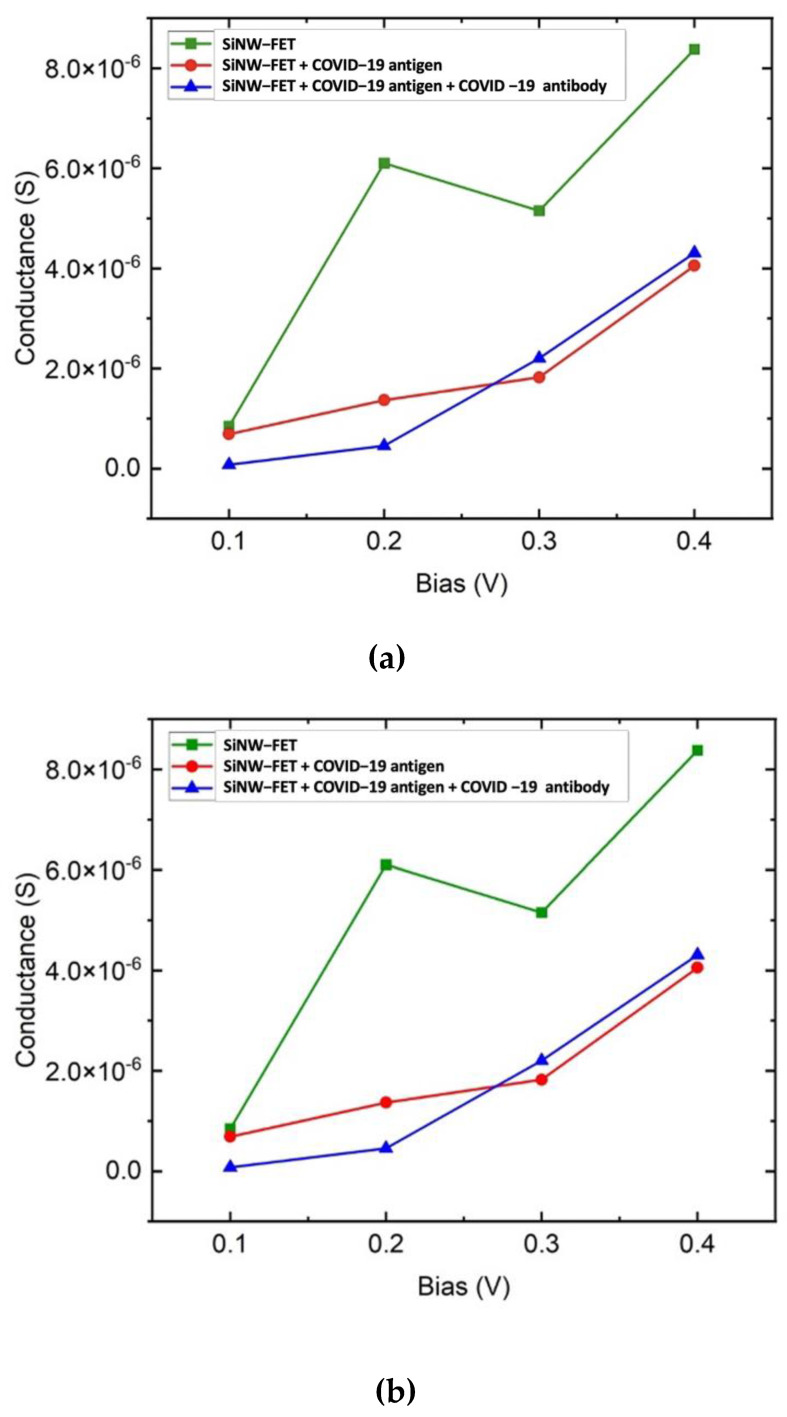
(**a**) Conductance and (**b**) IV-characteristics versus bias for the SiNW-FET (green), for the SiNW-FET with COVID-19 spike protein (red), and for the SiNW-FET with COVID-19 spike protein bound to COVID-19 antibody (blue).

**Figure 8 nanomaterials-12-02638-f008:**
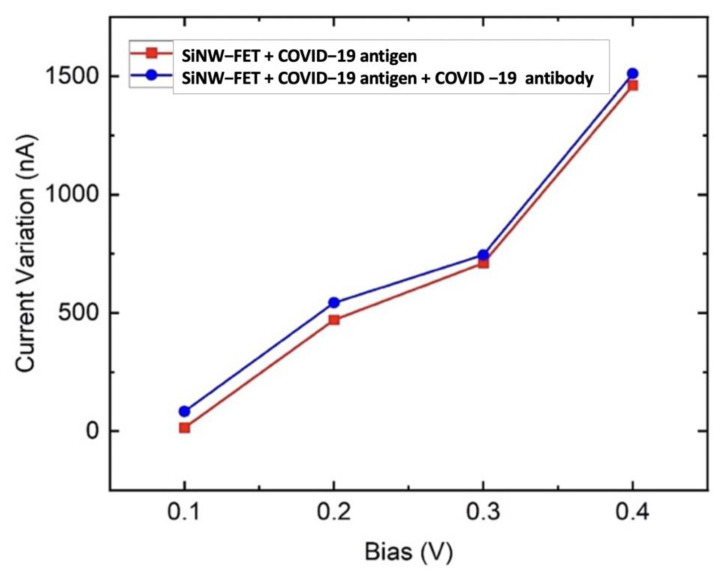
Variations in the electrical drain current due to the COVID-19 spike protein.

**Figure 9 nanomaterials-12-02638-f009:**
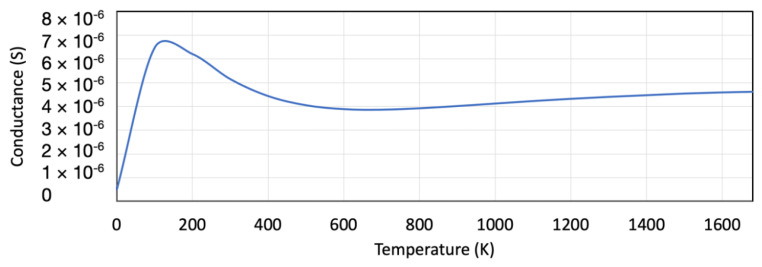
The temperature-dependent conductance for the SiNW field-effect transistor.

**Figure 10 nanomaterials-12-02638-f010:**
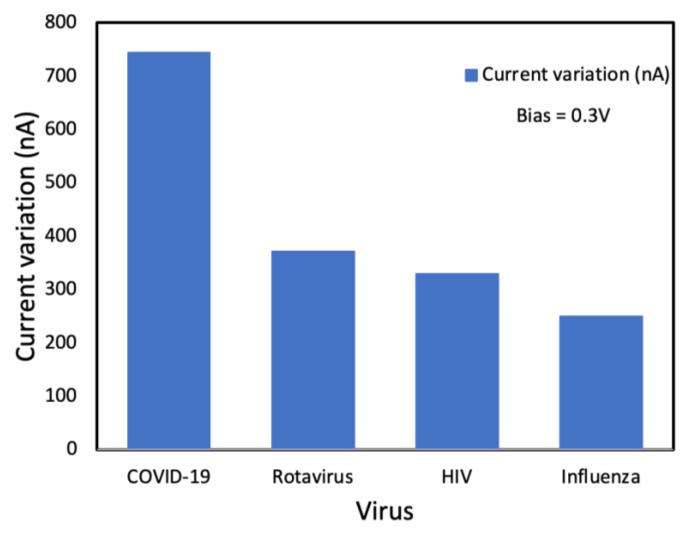
Change in the electrical drain current for different types of viruses.

## Data Availability

All data generated or analyzed during this study are included in this published article.

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
