# Peer review of "COVID-19 Detection via Silicon Nanowire Field-Effect Transistor: Setup and Modeling of Its Function"

_nanomaterials, 2022, doi:10.3390/nano12152638_

Round 1
Reviewer 1 Report
This paper seems to be helpful for semi-empirical modeling for COVID-19 detection using SN-FET. However, reviewer unfortunately could not find any technical and scientific meanings as compared with other researches in many published papers. Therefore, We easily could find these techniques to have high sensitivity, label-free, and specific detection as already published in many scientific journals.
1. For practical uses, SiNW-FETs can be carefully designed with various designs (lengths, widths, and channel geometries) for electrical properties of various FETs. Fabrication of SiNW-FETs issue is also critical for detection of COVID-19.
2. Chemical Modification methods of the SiNW-FETs to immobilize biomolecules such as spike proteins and corresponding antibodies also important which affect SiNW-FETs performances.
Author Response
Manuscript title: Semi-empirical Modeling for COVID-19 Detection via Silicon Nanowire Field Effect Transistor.
Manuscript number: Nanomaterials-1777610
Dear Editor,
The response to the reviewer comments is as below.
Response to the comments:
- Reviewer #1:
This paper seems to be helpful for semi-empirical modeling for COVID-19 detection using SiNW-FET. However, reviewer unfortunately could not find any technical and scientific meanings as compared with other researches in many published papers. Therefore, We easily could find these techniques to have high sensitivity, label-free, and specific detection as already published in many scientific journals.
Response: We thank the referee for his feedback on our manuscript. Definitely, the new version of manuscript has been enhanced based on the insights and guidance of the reviewers.
1: For practical uses, SiNW-FETs can be carefully designed with various designs (lengths, widths, and channel geometries) for electrical properties of various FETs. Fabrication of SiNW-FETs issue is also critical for detection of COVID-19.
Thanks for the comment.
Addressed, and below text was amended in page 3 in the revised manuscript.
The gate was made of two layers: a dielectric layer of SiO2 with relative dielectric K=3.9 and a metallic layer. The SiNW-FET channel has an approximate width of 57 Å and length of 75 Å. These dimensions were selected due to the number of atoms and the size of the COVID-19 target molecule. A pair of 12 Å electrodes was connected at the edges of the channel.
Moreover, it is worth noting that QuantumATK provide computational details for nanoscale sensors which can be used as a proof of concept to support the experiment which is based on bigger size sensors.
2: Chemical Modification methods of the SiNW-FETs to immobilize biomolecules such as spike proteins and corresponding antibodies also important which affect SiNW-FETs performances.
Thanks for the comment.
Addressed, and below text was amended in page 3 in the revised manuscript.
This study is a proof of concept that the designed SiNW-FET can be used as a sensor for COVID-19 virus detection. This work was able to find a unique electronic signature and detect COVID-19 antigen using the designed sensor.
The novelty in this article is based on the use of the SiNW-FET for the first time as a sensor to detect COVID-19 spike antigen. Moreover, semi-empirical model was used which is an extension of the extended Hückel approach with a self-consistent Hartree potential. Semi-empirical model has the advantage of lower computational cost and the possibility to use it in parallel with experiment where the model can be fitted to get accurate results [45].

Reviewer 2 Report
This manuscript offers a concise yet compelling report on a semi-empirical modeling approach to COVID-19 detection via silicon nanowire field effect transistor. The emphasis of the study falls on both sensor setup and configuration scheme and on the corresponding tailored modeling methodology fitted to GW calculations. A well-planed set-up is employed and masterfully chosen/adapted for the purpose. The modelling approach is also quite adequate for the task. The nice comparative context of results throughout the study strongly contributes to reliable scrutinization of COVID-19 detection via the silicone nanowire field effect transistor. From practical point of view, the reported results thus bring new knowledge and certainly represent an original contribution in the present context.
The authors chose an adequate structure of the manuscript – an excellent point of departure for such a study. Finally, the authors provided a balanced realistic and nicely illustrated presentation of their results and corresponding analysis that is of much scientific and practical interest and adds new knowledge to the field.
In my opinion, the fine detailing in the present work, the insightful and balanced discussion of the results, as well as the very good figures, permit competent readers to utilize the manuscript as a guidance for future work. Consequently, this manuscript presents an efficient and beneficial basis for promoting and solving next step challenges in this field.
The manuscript also benefits from a clear motivation, and it is an easy and informative read.
The present manuscript is a significant contribution, this work once published would be quite useful as well as instructive and suggestive in terms of further studies and to a wider readership.
There are some minor issues with this already excellent manuscript that will need to be addressed before becoming suitable for publication, i.e., it can be considered for publication after a minor revision:
1: Title can be optimized and made more attractive, something like: “COVID-19 detection via silicon nanowire field effect transistor: set up and modeling of its function.
2: In the introduction, the authors partly miss aspects of the general scope of modeling of electronic properties including excited state properties of silicon-based nanostructures for electronic applications. Examples for such advanced methodology applied to nanostructured materials of similar complexity include The Journal of Physical Chemistry C 118 (2014) 5501-5509; Chemical Physics Letters 458 (2008) 170-174. Such works illustrate support theoretical approach to realistic functional modelling by ab initio methods of similar nanostructured types of materials.
3: The authors mention room temperature as working temperature of the sensor which is fair enough. Still, temperature ranges related to thermal stability of the set-up should be discussed in the text.
4: A more explicit discussion of the structural and bonding aspects of the nanowire set-up in the text from the beginning maybe helpful to the reader for understanding the basic aspects of this set-up..
5: Spell-check and stylistic revision of the paper are still necessary. Some long sentences, misspellings, etc., still are noticeable throughout the text.
Author Response
Manuscript title: Semi-empirical Modeling for COVID-19 Detection via Silicon Nanowire Field Effect Transistor.
Manuscript number: Nanomaterials-1777610
Dear Editor,
The response to the reviewer comments is as below.
Response to the comments:
- Reviewer #2:
Response: We thank the Referee for the good impression on our revised manuscript.
1: Title can be optimized and made more attractive, something like: “COVID-19 detection via silicon nanowire field effect transistor: set up and modeling of its function.
Response:
Thanks for the comment. Addressed, the title was updated in page 1 in the revised manuscript.
2: In the introduction, the authors partly miss aspects of the general scope of modeling of electronic properties including excited state properties of silicon-based nanostructures for electronic applications. Examples for such advanced methodology applied to nanostructured materials of similar complexity include The Journal of Physical Chemistry C 118 (2014) 5501-5509; Chemical Physics Letters 458 (2008) 170-174. Such works illustrate support theoretical approach to realistic functional modelling by ab initio methods of similar nanostructured types of materials.
Response: We thank the Referee for pointing out these 2 references which further enriched our list of references. They have all been included in the list of references of our newly-revised manuscript. The new paragraph is added in page 2 in the revised manuscript:
Silicon is vastly utilized in modern microelectronic devices. Various theoretical and experimental works were carried out to understand the properties and the structures of silicon clusters doped with transition-metals (TM) [32,33]. These clusters are useful for solar cells, lithium-ion batteries, silicon-based catalysts, and cluster assembled materials. Single TM atom doped silicon clusters have special geometric structures, novel electronic properties, and special magnetic characteristics such as ferromagnetism and anti-ferromagnetism. The electronic properties can be modified by varying the central metal atom [32,33].
3: The authors mention room temperature as working temperature of the sensor which is fair enough. Still, temperature ranges related to thermal stability of the set-up should be discussed in the text.
Thanks for the comment.
Addressed, and below text was amended in page 10 and page 11 in the revised manuscript.
The field effect transistor temperature dependent conductance displayed in Figure 9 shows an initial decrease in conductance as a function of temperature. The conductance increases at the Fermi level due to the electron tunneling. At higher temperatures, the conductance increment starts stabilizing. Figure 9 shows that at 300 K and higher the FET conductance was stable.
Figure 9: The temperature dependent conductance for the SiNW field effect transistor.
4: A more explicit discussion of the structural and bonding aspects of the nanowire set-up in the text from the beginning maybe helpful to the reader for understanding the basic aspects of this set-up.
Thanks for the comment.
Addressed, and below text was amended in page 3 in the revised manuscript.
For the simulation, Si crystal [111] in the orientation of the nanowire was used. All atomic locations and the lattice constant are optimized. Reactive dangling bonds on silicon atoms are passivated with hydrogen to enhance the material’s stability and lifetime [47]. These dangling bonds can modify the bandgap energy of the material which affect its semiconductor properties.
5: Spell-check and stylistic revision of the paper are still necessary. Some long sentences, misspellings, etc., still are noticeable throughout the text.
Thanks for the comment.
The language of the manuscript has been revised and fixed accordingly
Reviewer 3 Report
In their paper "Semi-empirical Modeling for COVID-19 Detection ...", Asma Wasfi and coworkers present a computational study of the response of a SiNW FET to a COVID-19 antigen and antibody.
The research methodology is sound and the results obtained provide insights into the current variations due to absorption of antigen and antibody. The response is briefly compared to the effect of other viruses, which is of important for biosensor applications. This certainly makes the topic interesting for the readers of nanomaterials.
However, before the paper is ready for publication, the authors should address a few points:
1. line 102: what material do the authors have in mind when they choose a dielectric constant of 3.9? Does the type of metal not need to be specified?
2. page 2 and line 164: The authors should comment on how and to what extent the orientation of the protein and of the antibody on the SiNW affects the response of the FET.
3. line 148: What is actually the computational time?
4. line 153: Is E_f really the energy window width around the Fermi energy?
5. line 175-176: What are the A, B, and C directions?
6. line 186: What is the concentration of the antibody on the SiNW? A single one per NW? More? What effect does the concentration of antibody and antigen have on the FET response?
7. line 208: "T(E)" should be written without the quotation marks.
8. line 224: "The difference in the ..." - recheck this sentence.
9. Figure 6: Does transmission have no unit?
10. Figure 7: Can the authors introduce something like confidence intervals into the plot? Otherwise, it is not clear whether, for example, the dip in conductance at 0.3 V bias has a physical significance or not. The same applies to Figures 8 and 9.
11. Lines 287 - 294: The authors should clarify whether only the charge of antibody and antigen leads to a change in the response of the FET, or whether this also depends on the adsorption geometry.
I recommend that the paper should be reconsidered after major revision.
Author Response
Manuscript title: Semi-empirical Modeling for COVID-19 Detection via Silicon Nanowire Field Effect Transistor.
Manuscript number: Nanomaterials-1777610
Dear Editor,
The response to the reviewer comments is as below.
Response to the comments:
- Reviewer #3:
The reviewer does not point out all but only point out in part.
Response: We thank the referee for his feedback on our manuscript. The updated version of the manuscript has been enhanced under the recommendation of the referee.
[Methods]
1: no detailed information about COVID-19 antigen and antibody
- Simulation using antibody-antigen binding structure or physically adhesion model
Thanks for the comment.
Addressed, and below text was amended in page 5 in the revised manuscript.
In simulation antibody-antigen binding structure was utilized. The PDB ID of (Spike Protein) is 2IEQ [48] while the PDB ID of (COVID-19 virus spike receptor-binding domain complexed with a neutralizing antibody) is 7BZ5 [49]. The second structure consists of neutralizing antibodies binding to spike glycoprotein receptor of COVID-19 virus.
2: no information about the protein structures of HIV and rotavirus
- no PDB ID or references
- whole virus, spike protein, or others
Thanks for the comment.
Addressed, and below text was amended in page 5 in the revised manuscript.
The SiNW-FET selectivity was studied where the whole virus was used for HIV, rotavirus, and Influenza. The PDB ID for each of the viruses is as the following: for HIV is 4XFZ [50], for rotavirus is 3MIW [51], and for Influenza virus is 1EA3 [52].
3: no information about the simulation for HIV and rotavirus
- antibody exists or not
- antibody binds to other virus or not (should not bind)
Thanks for the comment.
Addressed, and below text was amended in page 5 in the revised manuscript.
This study is a proof of concept that the designed SiNW-FET can be used as a sensor for COVID-19 virus detection. Antibody was only used with COVID-19 spike target and it wasn’t added for the other viruses such as HIV and rotavirus due to the complexity of calculations and time required. The results indicate that the sensor shows the highest variation in current due to COVID-19 spike protein antigen in comparison to other types of viruses such as HIV, rotavirus, and Influenza virus. Moreover, the variation in current due to the binding between the spike protein antigen and antibody of COVID-19 virus was studied. In experiment, the COVID-19 spike antibody will be used to ensure the sensor selectivity to COVID-19 spike antigen [53-55]. It is expected that the target molecules of COVID-19 spike antigen will bind to the COVID-19 spike antibodies however the other viruses will not bind to the antibodies.
4: Current variation
- between what and what
Thanks for the comment.
Addressed, and below text was amended in page 9 in the revised manuscript.
Th SiNW-FET sensor was tested for different types of viruses. The variation in the current was used to evaluate the sensor performance. The change in the current was calculated as the current for the sensor due to the placement of target molecule minus the current of the bare sensor. The drop of the drain current of the SiNW-FET sensor after placing the target molecule is due to the negatively charged spike protein which induce excess hole carriers
[Results and graphs]
1 Figure 7(a)
- There is only explain what is Figure 7(a), and no further explanation.
- Why does conductance at 0.2 bias voltage for FET only yield high?
Thanks for the comment.
Addressed, and below text was amended in page 9 in the revised manuscript.
Figure 7 (a) reveals that the bare SiNW-FET have higher conductance than the SiNW-FET + Covid-19 antigen or the SiNW-FET + Covid-19 antibody + Covid-19 antigen. The addition of the antigen or antigen and antibody together modifies the surface of the SiNW-FET resulting in conductance reduction. The decrement in the SiNW-FET mobility occurred due to additional sources of dispersion of the charge carrier. This variation in conductance results in a unique signature due to the physical and chemical structures of the target molecule. COVID-19 spike protein is negatively charged which binds with the antibody resulting in a decrement in the conductance and current response. SiNW-FET shows maximum conductivity at 0.2 V. Thereafter, the conductivity gets reduced. The reduction in conductance at 0.3 V is due to negative differential resistance.
2 Current variation: If current variation in this manuscript means the difference between SiNW-FET and FET+antigen/FET+antigen
- The authors claimed that "Figure 254 8 shows that functionalizing the SiNW-FET channel with COVID-19 antibody results in higher variation in current and higher sensitivity to COVID-19 spike protein." But, the result is almost similar level. The authors should describe this point.
Thanks for the comment.
Addressed, and below text was amended in page 9 in the revised manuscript.
The spike antibody is essentially added to ensure the selectivity of the sensor to COVID-19 spike antigen. Also, it resulted in slight increment in current variation which means higher sensitivity. Thus, the addition of the COVID-19 spike antibody resulted in higher sensitivity and selectivity.
3 No results for FET+antibody only
Thanks for the comment.
Addressed, and below text was amended in page 9 in the revised manuscript.
Previous experimental work showed that functionalizing the transistor with antibody results in slight variation in current [44]. Thus, the simulation work didn’t include the results of SiNW FET + antibody only to avoid computational cost. The effect of functionalizing the sensor of antibody is expected to be small which results in slight changes in the current [44].
[Discussion]
1 Figure 9
- The authors did not discuss why they used bias voltage at 0.3 V.
- The selectivity is only 2-fold, which is not high enough selectivity.
Thanks for the comment.
Addressed, and below text was amended in page 10 in the revised manuscript.
In the designed sensor, the optimal results were generated when the bias potential is fixed among the left and right electrodes (+0.15 and −0.15 eV). Fixing the bias voltage at 0.3 V resulted in the best sensitivity readings.
One more virus is studied and figure 10 is updated accordingly.
Figure 10. Change in the electrical drain current for different types of viruses.
[Others]
1 There are numbering mistakes of subheadings.
Thanks for the comment.
The subheadings were checked and corrected accordingly.

Reviewer 4 Report
The reviewer does not point out all but only point out in part.
[Methods]
1 no detailed information about COVID-19 antigen and antibody
• Simulation using antibody-antigen binding structure or physically adhesion model
2 no information about the protein structures of HIV and rotavirus
• no PDB ID or references
• whole virus, spike protein, or others
3 no information about the simulation for HIV and rotavirus
• antibody exists or not
• antibody binds to other virus or not (should not bind)
4 Current variation
• between what and what
[Results and graphs]
1 Figure 7(a)
• There is only explain what is Figure 7(a), and no further explanation.
• Why does conductance at 0.2 bias voltage for FET only yield high?
2 Current variation: If current variation in this manuscript means the difference between SiNW-FET and FET+antigen/FET+antigen
• The authors claimed that "Figure 254 8 shows that functionalizing the SiNW-FET channel with COVID-19 antibody results in higher variation in current and higher sensitivity to COVID-19 spike protein." But, the result is almost similar level. The authors should describe this point.
3 No results for FET+antibody only
[Discussion]
1 Figure 9
• The authors did not discuss why they used bias voltage at 0.3 V.
• The selectivity is only 2-fold, which is not high enough selectivity.
[Others]
1 There are numbering mistakes of subheadings.
Author Response
- Reviewer #3:
The research methodology is sound and the results obtained provide insights into the current variations due to absorption of antigen and antibody. The response is briefly compared to the effect of other viruses, which is of important for biosensor applications. This certainly makes the topic interesting for the readers of nanomaterials.
Response: We thank the Referee for the good feedback on our manuscript. The updated manuscript has been improved under the guidance of the reviewers.
- line 102: what material do the authors have in mind when they choose a dielectric constant of 3.9? Does the type of metal not need to be specified?
Thanks for the comment.
Addressed, and below text was updated in page 3.
The gate was made of two layers: a dielectric layer of SiO2 with relative dielectric K=3.9 and a metallic layer. The metal layer can be made of palladium, platinum, iridium, or rhodium. However, ATK-VNL doesn’t provide the option to select the metal type.
- page 2 and line 164: The authors should comment on how and to what extent the orientation of the protein and of the antibody on the SiNW affects the response of the FET.
Thanks for the comment.
Addressed, and below text was updated in page 10.
The orientation effect of the protein and of the antibody is expected to be very little and shouldn’t affect the readings. Each molecule has a unique chemical and physical structure which affects the sensor current in a unique way resulting in a unique signature. This study is a proof of concept that the designed SiNW-FET can be used as a sensor for COVID-19 virus detection. The results were obtained by using the most common orientation where the antibody is on the channel surface and the antigen is bound to the antibody. This leads to a distinctive current for each type of virus.
- line 148: What is actually the computational time?
Thanks for the comment.
Addressed, and below text was updated in page 5.
Although a low concentration of the target molecules was used in simulation, each one of the IV curves took more than three weeks to be generated in HPC.
- line 153: Is E_f really the energy window width around the Fermi energy?
Thanks for the comment.
The text was checked and corrected in page 6 accordingly.
refers to the energy.
- line 175-176: What are the A, B, and C directions?
Thanks for the comment.
The below changes are updated in the manuscript in page 4 and page 6.
A, B, and C are indictors for A-, B- and C-direction as displayed in Figure 1 (b).
- line 186: What is the concentration of the antibody on the SiNW? A single one per NW? More? What effect does the concentration of antibody and antigen have on the FET response?
Thanks for the comment.
Addressed, and below text was amended in page 5.
The concentration of the antibody and antigen on the SiNW-FET has been kept low (one molecule) to reduce the required time to conduct the simulation. Although a low concentration of the target molecules was used in simulation, each one of the IV curves took more than three weeks to be generated in HPC. The current variation and sensitivity is expected to be more obvious with a higher concentration of the target molecules as shown in our previous work [53].
- line 208: "T(E)" should be written without the quotation marks.
Thanks for the comment.
The text was checked and corrected in page 8 accordingly.
- line 224: "The difference in the ..." - recheck this sentence.
Thanks for the comment.
The text was checked and corrected in page 8 as below:
The change in the transmission spectra due to the addition of COVID-19 spike protein has changed the total electric current of the SiNW-FET.
- Figure 6: Does transmission have no unit?
Thanks for the comment.
The transmittance is a ratio of intensity. Thus, the transmittance has no unit.
- Figure 7: Can the authors introduce something like confidence intervals into the plot? Otherwise, it is not clear whether, for example, the dip in conductance at 0.3 V bias has a physical significance or not. The same applies to Figures 8 and 9.
Thanks for the comment.
Acquiring confidence intervals would require many trials which is computationally expensive.
The text was updated in page 9 and page 10.
This study is a proof of concept that the designed SiNW-FET can be used as a sensor for COVID-19 virus detection. Each molecule has unique electronic structure which results in a significant electronic characteristic. The reduction in conductance at 0.3 V is due to negative differential resistance.
- Lines 287 - 294: The authors should clarify whether only the charge of antibody and antigen leads to a change in the response of the FET, or whether this also depends on the adsorption geometry.
Thanks for the comment.
The text was updated in page 10.
Both the negative charge of the antibody and antigen and the adsorption of the molecules (antibody and antigen) lead to a unique variation in the electrical current of the SiNW-FET. Each molecule has a unique electronic state, size, and interaction with the SiNW-FET channel. Each molecule density of states contribution at the Fermi level is unique due to the different spatial extension of the molecule.

Round 2
Reviewer 1 Report
This revised manuscript seems to be much improved for publication to the Nanomaterials now.
Author Response
Manuscript title: Semi-empirical Modeling for COVID-19 Detection via Silicon Nanowire Field Effect Transistor.
Manuscript number: Nanomaterials-1777610
Response to the comments:
- Reviewer #1:
Response: We thank the referee for his feedback on our manuscript. The referee comments has been followed to improve the manuscript.

Reviewer 3 Report
The authors have responded adequately to most of the concerns from the initial report. However, some questions still remain which should be answered by the authors before the paper is ready for publication.
1. What is plotted in Figure 6? Transmission as stated in the paper, or Transmittance as stated in the response letter? The authors should provide the definition to avoid any confusion.
2. Even so it is computationally expensive, the authors should state if or if not the nonlinear dependence of conductance vs. bias (Figure 7a) and current vs. voltage (Fig. 7b) has any physical meaning and how/if it can be rationalized.
3. Figure 9: Does it make any sense to plot conductance vs. temperature for temperatures larger than the melting point temperature of Si?
Author Response
Manuscript title: Semi-empirical Modeling for COVID-19 Detection via Silicon Nanowire Field Effect Transistor.
Manuscript number: Nanomaterials-1777610
Response to the comments:
- Reviewer #3:
Comments and Suggestions for Authors
The authors have responded adequately to most of the concerns from the initial report. However, some questions still remain which should be answered by the authors before the paper is ready for publication.
Response: We thank the referee for his valuable feedback on our manuscript. The referee comments and suggestions has been followed to enhance the manuscript.
- What is plotted in Figure 6? Transmission as stated in the paper, or Transmittance as stated in the response letter? The authors should provide the definition to avoid any confusion.
Thanks for the comment.
Addressed, and below text was corrected and updated in page 6.
The various electronic transport characteristics of the sensor can be generated, after obtaining the the self-consistent non-equilibrium density matrix. The transmission spectrum is one of the most notable transport properties of the system from which the current and conductance are calculated. The transmission coefficient T at electron energy can be obtained from the retarded Green's function [56].
The zero bias transmission spectrum between the source and drain was calculated using equation 1 [57,58]:
where, is the energy, is the trace, describes the broadening level because of the coupling to the electrodes, and are the self-energies presented by the electrodes.
- Even so it is computationally expensive, the authors should state if or if not the nonlinear dependence of conductance vs. bias (Figure 7a) and current vs. voltage (Fig. 7b) has any physical meaning and how/if it can be rationalized.
Thanks for the comment.
Addressed, and below text was amended in page 10.
Nonlinear resistance results in a nonlinear I-V curve. Examples of passive nonlinear devices are diodes, transistors, and thyristors. The nonlinear relation is due to using the semiconducting material of silicon which results in nonlinear resistance leading to nonlinear IV curve. Moreover, the reduction in conductance at 0.3 V is because of the negative differential resistance.
- Figure 9: Does it make any sense to plot conductance vs. temperature for temperatures larger than the melting point temperature of Si?
Thanks for the comment.
Addressed, and Figure 9 was updated in page 12.
Figure 9. The temperature dependent conductance for the SiNW field effect transistor.
